# Position: Stop Preaching and Start Practising Data Frugality for Responsible Development of AI

**Sophia N. Wilson** [1] **Andrew Millard** [2] **Guðrún Fjóla Guðmundsdóttir** [3] **Raghavendra Selvan** [1] **Sebastian Mair** [2]

## Abstract

This position paper argues that the machine learning community must move from preaching to practising data frugality for responsible artificial intelligence (AI) development. For too long, progress has been equated with ever-larger datasets, driving remarkable advances but now yielding increasingly diminishing performance gains alongside rising energy use and carbon emissions. While awareness of data frugal approaches has grown, their adoption has remained rhetorical, and data scaling continues to dominate development practice. We argue that this gap between preach and practice must be closed, as continued data scaling entails substantial and under-accounted environmental impacts. To ground our position, we provide indicative estimates of the energy use and carbon emissions associated with the downstream use of ImageNet-1K. We then present empirical evidence that data frugality is both practical and beneficial, demonstrating that subset selection methods can substantially reduce training energy consumption with little loss in accuracy, while also mitigating dataset bias. Finally, we outline actionable recommendations for moving data frugality from rhetorical preaching to concrete practice for responsible development of AI.

## 1. Introduction

Concerns about the environmental costs, social impacts, and power asymmetries of ever-larger datasets are increasingly voiced within the machine learning (ML) community as

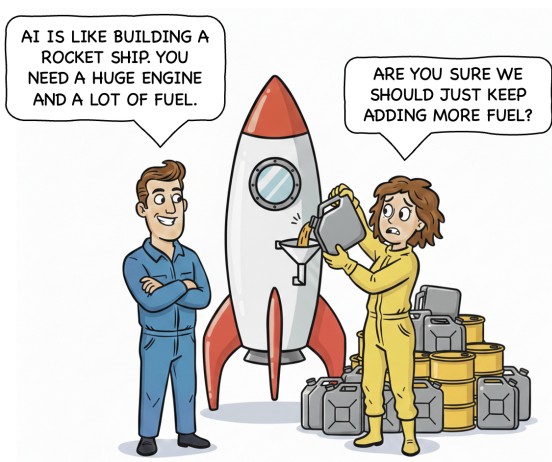

*Figure 1.* Illustration is inspired by a remark by prominent ML researcher Andrew Ng, comparing an AI model with a rocket ship (quoted in Kelly, 2016). However, progress in rocket science depends less on adding more fuel and more on using it wisely, reflecting the principle of data frugality. Created with Nano Banana (Google, 2026).

part of calls for responsible artificial intelligence (AI) development (Crawford, 2021). Parallel to these concerns, a growing body of work explicitly argues for scaling down rather than continuously scaling up (Goel et al., 2025; Wang et al., 2025). Yet, prevailing research incentives continue to reward scale, with reviewers, benchmarks, and leaderboards incentivising experiments conducted on increasingly large datasets (Ethayarajh & Jurafsky, 2020).

This is particularly evident in frontier AI development, where progress is often equated with ever larger datasets, formalised through empirical scaling laws (Kaplan et al., 2020; Hoffmann et al., 2022). Performance gains are, however, increasingly exhibiting diminishing returns, while computational demand and associated carbon emissions continue to rise (Strubell et al., 2020; Luccioni et al., 2023; Faiz et al., 2024). Moreover, large-scale datasets are known to contain substantial redundancy and noise (Lee et al., 2022; Penedo et al., 2023; Tirumala et al., 2023), raising questions about the efficiency of continued data expansion.

[1]Department of Computer Science, University of Copenhagen, Copenhagen, Denmark [2]Division of Statistics and Machine Learning, Linköping University, Linköping, Sweden [3]Department of Electrical and Photonics Engineering, Technical University of Denmark, Kongens Lyngby, Denmark. Correspondence to: Sophia N. Wilson <sophia.wilson@di.ku.dk>.

*Proceedings of the 43rd International Conference on Machine Learning*, Seoul, South Korea. PMLR 306, 2026. Copyright 2026 by the author(s).

A natural response to these issues is to recognise that not all data are equally informative (Katharopoulos & Fleuret, 2018) and to adopt data frugal approaches that maximise learning efficiency per data sample, reflecting the conceptual message of Figure 1. One prominent example is coreset selection, which constructs smaller yet representative datasets that preserve task performance while reducing training computation (Feldman, 2019). Despite often being preached as a way to reduce energy, many coreset methods do not report the accuracy per unit energy, nor the energy cost of subset construction itself, creating a gap between what is preached and what is practised. Considering a representative sample of ten coreset methods applicable to deep learning[1], the stated motivations for data reduction includes computational efficiency (8/10), energy efficiency (3/10), and reduced storage requirements (2/10). Two papers also mention the democratisation of AI, by enabling participation without access to large-scale computational resources. Despite this, only six papers report time savings and only one evaluates energy savings. The latter, would have been straightforwardly assessed using tools such as Carbontracker (Anthony et al., 2020) and CodeCarbon (Courty et al., 2024). These patterns motivate our central position:

**The ML community must move from preaching to practising data frugality for responsible development of AI.**

To operationalise this shift, we first provide background on the environmental impacts of data in ML, explain how data frugal approaches can limit these impacts, and situate our work within the broader literature (Section 2). We then quantify the aggregate energy use and associated carbon emissions arising from dataset use and storage, making explicit what data frugal approaches aim to reduce (Section 3). Next, we present empirical evidence that data frugality is both practical and beneficial (Section 4). We show that coreset selection can substantially reduce training energy consumption with little loss in accuracy, and that coreset-based dataset curation can mitigate dataset bias. We then critically discuss our position (Section 5) and outline actionable recommendations for how to practise data frugality (Section 6). Finally, we discuss counter-arguments to data frugality (Section 7) and summarise our position (Section 8).

## 2. Background

We treat data and models as having distinct lifecycles, as illustrated in Figure 2, which motivates a corresponding distinction between data frugality and model frugality, with the former being the focus in this paper.

[1]We do not enumerate individual papers, as our goal is to highlight systematic reporting patterns rather than critique specific contributions, but they can be provided upon request.

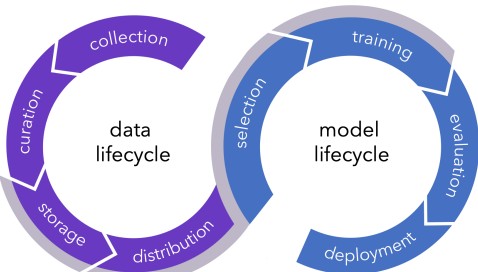

*Figure 2.* Visualisation of a data lifecycle (purple) and a model lifecycle (blue). Subset selection methods can have positive effects on multiple parts across both lifecycles (grey).

### 2.1. Environmental Costs of Data in ML

The environmental costs associated with data have long been overlooked. As a result, data are frequently hoarded (Veritas Technologies LLC, 2018), with more than half of all stored data never accessed or used after its creation (Splunk, 2019). A similar dynamic is evident in ML, where the growing creation and use of large-scale data has led to a form of hyper-datafication in which the associated costs are largely ignored (Wilson et al., 2026).

Within the ML community, concern about the environmental costs has largely centred on model development (Henderson et al., 2020) and deployment (Luccioni et al., 2024) rather than on the data itself. One exception is an analysis of the BLOOM large language model (LLM) by Luccioni et al. (2023), which explicitly accounts for several processes beyond training including data processing, tokenization, and benchmarking. These activities are estimated to contribute 3.3 tonnes of carbon dioxide equivalent ($tCO_2e$) emissions, corresponding to 5% of total emissions. An industry estimate by Common Crawl and Tailpipe[2] shows that crawling five billion web pages costs about 326 $kgCO_2e$, and that relocating the crawl to a lower-carbon grid could reduce emissions by up to 90%. Taken together, these examples show that data-related processes can represent a non-negligible share of overall system impacts, even when considered only partially. Yet explicit carbon accounting remains rare. Soldaini et al. (2024), for instance, report the compute infrastructure used to filter and deduplicate a 3-trillion-token pretraining corpus, though do not convert these costs to energy or carbon equivalents.

In practice, environmental costs incur across the full data lifecycle and include both embodied and operational impacts. Embodied impacts arise from the extraction of raw materials, manufacturing, transport, installation, and disposal of hardware and data centre infrastructure. Operational impacts stem from the electricity required to store, transmit, and process data, as well as from associated

[2]https://tailpipe.ai/measuring-the-carbon-cost-of-crawling-five-billion-web-pages/

resource demands such as water use for cooling and land use for data centre construction. The Common Crawl study reports that 2% of their emissions stem from embodied emissions and 98% from operational electricity use. Many of these impacts can, however, be difficult to attribute accurately. For example, network electricity use is often estimated using traffic-proportional averages, even though network energy consumption is largely driven by baseline capacity rather than marginal data transfer, which can lead to substantial overestimation (Mytton et al., 2024).

Despite their cumulative scale, data-related energy costs are diffusive and weakly attributed (Mersico et al., 2024), which makes them difficult to quantify. As a result, data are frequently treated as an effectively free input to ML models. Changing this requires making the environmental impacts of data visible across the data lifecycle and explicitly reporting energy metrics. Recent work has therefore proposed attaching the information about these costs directly to datasets, for example as meta-data covering each step of the data lifecycle (Mersy & Krishnan, 2024).

## 2.2. Data Frugality

The frugal mindset aims to maintain model performance while minimising resource consumption (Evchenko et al., 2021; Violos et al., 2026). In this work, we define **data frugality** as the practice of maximising learning efficiency per data sample by identifying and removing redundant or uninformative samples using subset selection methods.

For many learning problems, dataset size can be reduced substantially without noticeable drops in performance (He et al., 2024; Tan et al., 2025). This can be achieved by computing representative subsets, i.e., small subsets of large datasets that contain the most informative data samples (Feldman, 2019). Models trained on such a subset are expected to achieve performance comparable to those trained on full datasets. Subsets with approximation guarantees are often called coresets. Apart from improving storage demands and model training, those representative subsets are also beneficial for model selection.

Data frugality can be pursued through several complementary strategies. Dataset condensation and distillation synthesise compact training sets, while subset selection retains a subset of existing samples. This paper focuses on the latter and we use *subset selection* (or *data pruning*) as the umbrella term, and reserve *coreset* for subsets with formal approximation guarantees. Many state-of-the-art (SOTA) methods discussed here are score-based heuristics that retain accuracy empirically without such guarantees (see Appendix A).

By prioritising data quality over data quantity, frugal approaches can surpass classical power-law scaling (Sorscher et al., 2022), while reducing computational demand and environmental impact across multiple stages of the data and model lifecycle (see Figure 2) (Violos et al., 2026; Penedo et al., 2023; He et al., 2024; Verdecchia et al., 2022). In the LLM setting, Muennighoff et al. (2023) show that data-constrained training using repeating data tokens across epochs, achieves performance competitive with unconstrained training, suggesting that data frugality is viable even at frontier scale. Datasets that are small from the outset typically incur lower energy use across the entire data lifecycle, as well as model selection and training, compared to a larger dataset. Meanwhile, coreset methods primarily reduce energy during model training, and can also lower costs associated with storage, distribution, and model selection. Empirical work shows that coreset approaches can deliver substantial training energy reductions with limited performance loss (Killamsetty et al., 2021; Scala et al., 2025).

The benefits of data reduction extend beyond energy and carbon savings. Smaller datasets lower computational barriers, improving accessibility (Ahmed & Wahed, 2020; Rather et al., 2024), enhance reproducibility, facilitate representativeness (Clemmensen & Kjærsgaard, 2022), support privacy-preserving and resource-efficient training in distributed settings (Albaseer et al., 2021; Bian et al., 2024), and can mitigate ethical and representational harms associated with large-scale datasets (Birhane & Prabhu, 2021; Birhane et al., 2022).

## 2.3. Related Work

A growing body of recent position papers challenges the assumption that ever-larger datasets are the primary driver for progress in ML. Instead, they highlight diminishing returns, inefficiencies, and rising environmental impacts.

**The cost of data is often overlooked.** Kandpal & Raffel (2025) argue that while the growing computational, hardware, energy, and engineering demands of LLMs are frequently considered, the human labour underpinning training data is often overlooked. Books, papers, code, and online content created by humans form the foundation of LLM training corpora. They propose assigning monetary value to this labour and argue that it should constitute the most expensive component of LLM training. They estimate that the monetary value of training data already exceeds model training costs by one to three orders of magnitude. A similar perspective is offered by Jia et al. (2025), who analyse the economic data value chain and argue that value systematically accrues to data aggregators and model developers, while data generators remain largely uncompensated. They advocate for equitable data valuation, provenance, and compensation mechanisms. While these works focus on the economic valuation of data, we instead argue for a more frugal usage of it. Furthermore, Santy et al. (2025) argue

that current research incentives reward data volume over care, context, and human labour, leading to datasets that are large yet poor in signal. While their focus is on labour, dignity, and governance, their analysis reinforces our claim: treating data as an abundant, low-cost resource reinforces wasteful and misaligned practices. Data frugality can be read as a technical response that partially realigns incentives by rewarding careful selection over wasteful accumulation.

**Current LLM scaling proceeds in the wrong direction.** Wang et al. (2025) argue that scaling within AI development usually means scaling up (data, model parameters, compute), whereas future progress should emphasise scaling down or scaling out, including reductions in carbon footprint. They note that large-scale pretraining has already exhausted much of the publicly available high-quality web content and that the remaining content is either low-quality or AI generated. Thus, continued dataset expansion will no longer result in the same performance gains as earlier. As a remedy, they identify data-efficient training and using smaller, high-quality datasets as future directions. This perspective is also shared by Goel et al. (2025), who advocate downscaling datasets and LLMs to reduce resource demands while maintaining performance. One of their recommendations aligns with ours: adopting a more frugal approach to data.

**We are measuring the wrong things.** McCoy et al. (2025) explicitly challenge "scaling fundamentalism" and propose capability-per-resource as a more appropriate metric for AI progress. They argue for selective data usage and explicit resource-aware reporting as a way forward. Reporting practices are also central to the arguments of Wilder & Zhou (2025). They argue that research incentives shape behaviour and narrow evaluation metrics distort what gets optimised. Current benchmarks and leaderboards often incentivise data accumulation, making evaluation norms part of the sustainability problem. We largely agree with these critiques and provide concrete recommendations especially for research on data reduction techniques.

**AI should be more democratic.** Upadhyay et al. (2025) propose a regulatory mandate: frontier labs should release small, open "analogue models" distilled from their large proprietary systems. These analogues can enable research on safety, interpretability, and alignment without exposing full-scale models. While their proposal targets models, we make a complementary argument for smaller, highly informative datasets. Such datasets have lower storage costs, are easier to share, and enable training under limited computational resources.

**Our contribution.** Overall, these position papers converge on the view that continued reliance on scale alone is increasingly inefficient and unsustainable. Our contribution complements this literature by focusing on data frugality and explicitly addressing the gap between preach and practise.

Rather than limiting our contribution to critique and high-level recommendations, we make both dataset costs and the gains achievable through data frugality explicit and provide concrete guidance for translating these insights into practice.

## 3. Estimating the Energy and Carbon Costs

Estimating the energy and carbon costs of a dataset clarifies what data frugal approaches aim to reduce. In this section, we estimate the downstream energy use of ImageNet from model training and storage, the two lifecycle stages most directly affected by coreset selection.

ImageNet is one of the most influential datasets in computer vision (Deng et al., 2009). While the full dataset comprises over 14 million images and about 21,000 classes (ImageNet-21K), we focus on the most widely used subset, ImageNet-1K, which contains 1.28 million training images spanning 1,000 classes and occupying 130 GB of storage. ImageNet-1K served as the benchmark dataset for the ImageNet Large Scale Visual Recognition Challenge (ILSVRC) between 2010 and 2017 and remains a canonical training dataset today (Russakovsky et al., 2015).

### 3.1. Model Training

To estimate aggregate downstream model training, we first approximate the number of independent training runs performed on ImageNet-1K. We base this estimate on an analysis of all accepted ICLR papers from 2017 to 2022 retrieved via the OpenReview API. We focus on ICLR because, with the exception of 2016, all conference editions use OpenReview for peer review, providing consistent and comprehensive coverage over time. For each year, we estimate the fraction of ImageNet-mentioning papers that report training a model from random initialisation, as opposed to using ImageNet for fine-tuning, evaluation or as a reference. We then extrapolate this fraction to 2023–2025 using linear regression. Assuming this fraction is representative beyond ICLR, we apply it to the broader literature by combining it with a keyword-based count of publications mentioning *imagenet* using dimensions.ai[3]. This procedure yields an estimated 46,179±1,154 training runs on ImageNet-1K between 2017 and 2025, where the uncertainty reflects the empirical error rate of 2.5% from the LLM-based labelling (Appendix B). The estimates neglect training prior to 2017, however, the usage during the ILSVRC involved a relatively small number of competing teams compared to the scale of subsequent academic use (Russakovsky et al., 2015).

We estimate the energy consumption of a single training run using a standard reference setup for ImageNet training[4]. Specifically, we consider training a ResNet-50 model

---

[3]https://app.dimensions.ai
[4]https://github.com/pytorch/examples/tree/main/imagenet

on a single NVIDIA A100 GPU within a DGX A100 machine[5] and adopt 300 epochs as a representative training regime (Wightman et al., 2021). We measure energy consumption using Carbontracker[6] (Anthony et al., 2020). Under this setup, the average energy use is estimated at 0.394 kWh per training epoch. See Appendix B for details.

The total energy consumption associated with ImageNet-1K training therefore amounts to:

$$(46,179 \pm 1,154) \times 0.394 \text{ kWh/epoch} \times 300 \text{ epochs}$$
$$= 5.46 \pm 0.14 \text{ GWh}.$$

Using a global average carbon intensity of 445 gCO$_2$e/kWh (IEA, 2025), this corresponds to approximately 2429±61 tCO$_2$e, equivalent to the annual carbon footprint of around 514±13 people based on a global per-capita average of 4.73 tCO$_2$e (Ritchie et al., 2023).

### 3.2. Data Storage

We estimate storage-related energy consumption by assuming that each training run corresponds to one retained local copy of ImageNet-1K. Using a dataset size of 130 GB and an annual energy intensity of 60 kWh/TB/year (Selvan, 2025), the energy required to store these copies for one year is:

$$(46,179 \pm 1,154) \times 130 \text{ GB} \times 60 \text{ kWh/TB/year}$$
$$= 360 \pm 9 \text{ MWh}.$$

This corresponds to approximately 160±4 tCO$_2$e, or the annual carbon footprint of around 34±1 people.

Carbon footprint estimates are highly sensitive to the carbon intensity of the local energy grid, which varies by more than a factor 60 across countries (Ember & Energy Institute, 2025). To illustrate this sensitivity, we convert the total energy consumption of $5.82 \pm 0.15$ GWh to carbon footprints under three scenarios: using the global average (445 gCO$_2$e/kWh), a high-carbon grid such as Turkmenistan (1,310 gCO$_2$e/kWh), and a low-carbon grid such as Lesotho (21 gCO$_2$e/kWh). This yields estimates of $2589 \pm 65$ tCO$_2$e, 7624±191 tCO$_2$e, and 122±3 tCO$_2$e, respectively.

Importantly, the estimates for both model training and storage represent strict lower bounds. As of 1 December 2025, the Hugging Face Hub alone stored 876 versions, copies, and derivatives of ImageNet[7], as shown in Figure 7 in Appendix B. Of these, 214 explicitly include *imagenet-1k* in their dataset identifier. Summing the downloads of these 214 datasets alone yields over 2.5 million downloads, a factor 55 larger than our estimate of distinct training runs. This indicates that the true scale of ImageNet-1K usage substantially exceeds what is captured by publication-based counts.

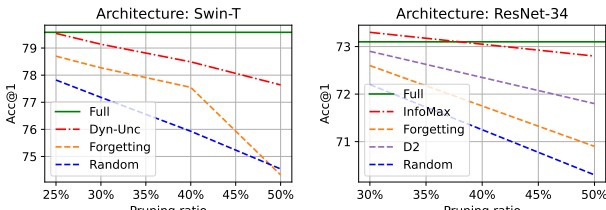

*Figure 3.* Top-1 accuracy of pruning ImageNet-1K using Dyn-Unc (left), D2 (right), and InfoMax (right). Forgetting, Random, and the performance on the full ImageNet-1K are shown as references. The numbers are taken from He et al. (2024) and Tan et al. (2025). Note that the architectures differ, there is no performance loss for 25%–30% of data pruning when using Dyn-Unc/InfoMax, and that the authors do not compare their methods against each other.

However, even under these conservative assumptions, downstream use of ImageNet-1K alone entails substantial energy consumption and associated carbon emissions, comparable to the annual emissions of hundreds of individuals, underscoring that dataset-level impacts warrant explicit consideration in environmental assessments of AI systems.

## 4. Gains from Data Frugal Approaches

We now show that subset selection can provide positive effects on time and energy demands of model training as well as on mitigating dataset biases without sacrificing accuracy[8].

### 4.1. Pruning Effects on Accuracy

Following our analysis on the energy use and carbon emissions associated with downstream use of ImageNet-1K, we next report SOTA results for subset selection techniques on this specific dataset. Figure 3 presents results across multiple pruning ratios, evaluated on two architectures: Swin Transformer (Liu et al., 2021) and ResNet-34. The results for Swin-T are from He et al. (2024), who propose the subset selection method Dyn-Unc while the results for ResNet-34 are from Tan et al. (2025), who introduce the subset selection method InfoMax. Note that He et al. (2024) and Tan et al. (2025) do not compare their approaches against each other and use different architectures. To put the data pruning results into perspective, we also report on the performance that can be achieved using the full data set (0% pruning) and using a subset sampled uniformly at random. In addition, the performance of the pruning methods forgetting scores (Toneva et al., 2019) and $\mathbb{D}^2$ pruning (Maharana et al.) is provided.

As seen in Figure 3, all subset selection methods perform better than a random subset (blue line) as they yield higher accuracies per pruning ratio. Notably, Dyn-Unc (He et al., 2024) (red line, left plot) can prune 25% of both ImageNet-1K and ImageNet-21K (not shown) without

---

[5]https://docs.nvidia.com/dgx/dgxa100-user-guide

[6]https://github.com/saintslab/carbontracker

[7]https://huggingface.co/datasets?sort=trending&search=imagenet

[8]Code available at https://github.com/saintslab/data-frugality

*Table 1.* Training time (minutes) and energy consumption (kWh) per epoch on full ImageNet-1K and a 25% pruned subset sampled uniformly at random for various architectures using a single NVIDIA A100 GPU. The estimates are averaged over 30 epochs.

| Model | Param. | minutes/epoch | | | kWh/epoch | | |
|---|---|---|---|---|---|---|---|
| | | Full | 25% pruned | Gain | Full | 25% pruned | Gain |
| ResNet-34 | 21.8M | 35.2 | 23.8 | 32% | 0.2798 | 0.1989 | 29% |
| ResNet-50 | 25.6M | 40.7 | 24.3 | 40% | 0.3940 | 0.2645 | 33% |
| Swin-T | 28.3M | 58.7 | 44.6 | 24% | 0.7002 | 0.5300 | 24% |

any drop in Top-1 accuracy (compare against green line). Similar results can be observed for InfoMax (Tan et al., 2025) (red line, right plot), which can prune up to 35% of ImageNet-1K without a sacrifice in performance. This shows that large datasets such as ImageNet-1K can be pruned with no to little drops in performance.

### 4.2. Pruning Effects on Time and Energy Demands

Next, we train classifiers based on ResNet-34, ResNet-50, and Swin-T architectures on ImageNet-1K three times for ten epochs using the same experimental setup as in Section 3. Our goal is to estimate the computation time and energy consumption (CPU and GPU) per epoch for the full and a pruned version. Table 1 depicts the results. Pruning 25% of the dataset reduces approximately 32% of the training time and 29% of the energy consumption per epoch when using a ResNet-34 backbone. Note that these substantial reductions are possible without noticeable performance losses using SOTA data reduction techniques (see Figure 3). One caveat is that we neglect the computational burden of constructing the coreset. However, this is a one-time cost which we discuss in Section 4.4. In addition to the benefits of model training, a 25% dataset reduction has positive effects on the energy required for storage.

### 4.3. Bias and Representation Effects

Data frugality principles urge us to choose a smaller, representative subset from a larger dataset. As seen above, this can offer a compelling trade-off between performance and energy consumption. Coupling coreset curation with other objectives than dataset coverage can improve other properties of the dataset (Chen & Selvan, 2025). For instance, any model that is trained on a biased dataset, wherein a single majority group is over-represented compared to other minority groups, can also learn to reproduce these biases in its predictions (O'Neil, 2016). Coresets can be selected to mitigate these biases by balancing the samples between groups so that no single group is over-represented at the expense of other groups (Barocas et al., 2023). This is particularly effective if the data collection itself cannot be corrected due to systemic problems and/ or historical reasons (Wilson et al., 2026). Adjusting the data distribution by reweighting minority samples or subsampling majority groups is a

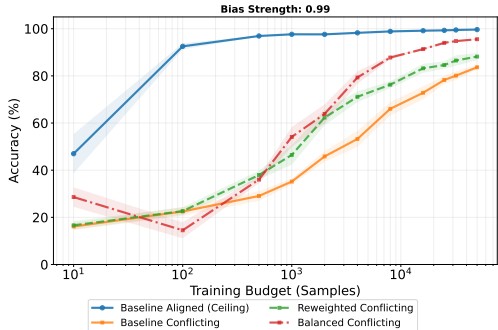

*Figure 4.* Performance of a classifier trained on the biased Colour-MNIST (99% majority group) dataset is reported for the case with no bias (aligned) and with bias (conflicting). Three coreset methods are shown: random (baseline), reweighted, and balanced.

well-established approach to mitigating bias (Kamiran & Calders, 2012; Krasanakis et al., 2018), and coresets offer an algorithmic remedy to mitigate known biases.

We illustrate this using a toy example based on a colourised version of the MNIST dataset[9]. We associate each digit with a majority colour, while the remaining samples are randomly coloured, as shown in Figure 8 in Appendix C. An ideal classifier should not focus on the colour of the digits, but instead should focus on other relevant features like shape.

Figure 4 shows the influence of different data budgets on classification performance. We report the *aligned* accuracy which is the best accuracy possible if the models do not focus on colours, whereas the *conflicted* accuracy is shown for the case where 99% of samples belong to a majority group and 1% of the samples belong to the minority group. The baseline method is random sampling, which is compared with two other data sampling methods. In the reweighted sampling method, the loss of data points are reweighted inversely proportionally to their groups, i.e., majority group samples are under-weighted. In the balanced sampling, the number of samples between the majority and minority groups are rebalanced so that the bias is removed within the coreset. In budget regimes where the minority samples are fewer than the required budget, we augment the data samples. The trends in Figure 4 capture the usefulness of curating coresets by altering the underlying data distribution which can remove known biases in the dataset. See additional results for different bias strengths in Appendix C.

While our experiment uses a controlled toy setting, recent work on fair coreset selection and dataset condensation demonstrates that bias-aware data curation can simultaneously improve accuracy and fairness on more realistic and complex datasets (Zhang et al., 2026; Zhou et al., 2025), further supporting the generalisability of the approach.

---

[9]See https://github.com/kakaoenterprise/Learning-Debiased-Disentangled and https://ieee-dataport.org/documents/colored-mnist

## 4.4. Amortisation of Coreset Construction Costs

Constructing a representative subset often incurs an upfront one-time cost that varies considerably across methods. At one extreme, random uniform pruning requires no construction overhead. At the other, methods such as Dyn-Unc may require computation equivalent to a full training run on the entire dataset. In the latter case, adoption is only justified if the construction cost is amortised across subsequent uses, for example, through repeated use in model selection or as a shared dataset that reduces downstream storage, distribution, and training costs for every user.

To make this concrete, we consider the scenario of an expensive coreset method whose construction cost is equivalent to one full training run on the entire dataset in the context of our ImageNet case study. Training on a 25% pruned subset saves 24–33% energy (Table 1), meaning that the construction cost is amortised after three to four training runs. Had a frugal version of ImageNet-1K been constructed and shared from the outset, the one-time construction cost would have been negligible relative to the aggregate savings. Concretely, a 25% reduction in dataset size would have saved approximately 1.4–1.9 GWh, corresponding to around 621–854 $tCO_2e$ (using the global average carbon intensity at 445 $gCO_2e$/kWh), even under our conservative lower-bound assumptions.

## 5. Discussion

**Disentangling compute, energy, and carbon efficiency.** A common misconception is that improvements in compute efficiency directly translate to energy efficiency, and that gains in energy efficiency in turn imply proportional reductions in carbon emissions. However, this is not necessarily true, see, for example Wright et al. (2025). This distinction is also evident in our experimental results (Table 1, Section 4.2). Reducing the size of ImageNet-1K by 25% does not yield a corresponding 25% reduction in time or energy use. Instead, we observe time reductions ranging from 24–40% and energy reductions from 24–33%. These results illustrate that dataset size, training time, and energy consumption cannot be treated as interchangeable proxies. Assuming proportional reductions based solely on data pruning is therefore misleading. Time and energy must be measured and reported explicitly if efficiency claims are to be meaningful.

**Alternatives to data frugality.** Alternatives to data frugality emphasise efficiency gains at the level of models or hardware rather than datasets. Advances in model architectures, training algorithms, hardware acceleration, and scheduling can mitigate some of the environmental impacts without constraining the amount of data used, thereby preserving the benefits of large and diverse datasets. In practice, these interventions can be easier to adopt because they can be implemented unilaterally, whereas using smaller datasets can conflict with comparability and benchmarking.

**Closing the gap.** Across the literature, efficiency and energy considerations are frequently invoked to motivate data reduction, yet the corresponding metrics are often not reported (as exemplified in Section 1). This gap is not technical, as energy consumption and associated carbon emissions can be straightforwardly tracked using lightweight tools such as Carbontracker (Anthony et al., 2020) and CodeCarbon (Courty et al., 2024). Making such reporting routine would be a first practical step towards aligning stated motivations with actual practice.

Quantifying the full environmental impacts of the full data lifecycle is challenging and requires development of new tools and standards. Dataset creators mention limited resources, including computational constraints, as a challenge during dataset creation (Orr & Crawford, 2024). When estimates of the compute used in dataset creation are available, reporting them would therefore be a natural extension of existing documentation and would improve transparency without imposing substantial additional burden.

Finally, concerns about comparability and reporting burden are not unique to energy and carbon metrics. Similar arguments have long been used in debates around code release and runtime in benchmarking. Acknowledging these challenges is important, but they should not preclude progress. Encouraging incremental transparency through reporting and community norms is a necessary step towards practising responsible AI development.

**Costs, limitations and open challenges of coresets.** While coreset methods can reduce computational costs, these gains come with trade-offs. Coreset construction introduces additional computational overhead and does not reduce impacts incurred during data collection or curation. This upfront cost is only amortised through reuse, limiting its effectiveness in one-off training scenarios. Moreover, removing data points can have unintended consequences that can be difficult to assess in advance (see Section 7). Coreset methods also face methodological limitations. Most are designed for a specific objective, such as classification losses or gradient alignment (see Appendix A). This is often sufficient for discriminative learning tasks such as ImageNet classification but less suitable for generative modelling, where preserving fine-grained structure, rare modes, and long-tail variation is critical. As a result, a subset that preserves classification accuracy may be inadequate for training generative models such as diffusion models, although, research in that direction exists (Chen & Selvan, 2025). More generally, coreset quality can be sensitive to architectural choices (see Section 4.1), optimisation settings, and data augmentations, and aggressive data pruning may amplify bias.

**Limitations of our paper.** Our estimates in Section 3 omit unpublished training runs, broader lifecycle impacts beyond electricity use, and ImageNet-related usage beyond the ImageNet-1K subset. The scope and limitations of these estimates are further elaborated in Appendix B. Moreover, all experiments focus on image datasets and the results may not generalise for tokenised or multimodal datasets used in frontier models.

## 6. Call to Action

Our position identifies the value-action gap between the preaching and practising of data frugality. Such value-action gap is a common occurrence in activism and has been studied extensively in the context of climate action (Blake, 1999). There are several reasons for the value-action gap to persist; primary of these is the *relatively* low effort required to ideate compared to tangible action which can also be limited due to external constraints. We next present a call to action that aims to *incentivise* relevant stake-holders to pursue data frugality for the responsible development of AI.

**People.** Individuals have the most immediate agency to challenge the status quo and shift practice away from data scaling by adopting data frugality in their own research choices and workflows.
○ *Measure and report data frugality:* Report resource consumption (and savings) due to data frugality to increase awareness of data-related costs, alongside model-related costs.
○ *Performance per data-point as metric of success:* Transition from aggregate performance to "Data-Pareto" reporting. This shifts the focus from performance-only to performance relative to dataset size or individual dataset contribution.
○ *Share data with care:* Dataset creators should consider curating frugal datasets before publishing them by removing redundant samples. Applying coreset methods at creation time can reduce wasteful storage, transmission, and computation. Energy and carbon costs incurred during collection and curation should be documented as part of dataset metadata, following the spirit of Gebru et al. (2021).
○ *Measure what you motivate:* When practitioners motivate their approach with something that is measurable, for example energy consumption, it should be evaluated.

**Platforms.** Data storage and benchmarking platforms, ML communities, and conferences can encourage and reward data frugality by shaping everyday research practice through submission requirements, leaderboards, and recognition mechanisms.
○ *Make resource reporting easy and mandatory:* Adopt mechanisms such as CVPR's mandatory compute reporting form and accompanying awards for recognition[10].

○ *Resource-aware benchmarking:* Maintain canonical benchmark datasets with streamed access and persistent leaderboards to reduce redundant downloads, storage, and repeated baseline training. This should be complemented by metrics such as kWh to generalise across hardware and time.
○ *Incentivise data-efficient research:* Host challenges such as Energy Efficient Image Processing[11] or BabyLM[12] that seek and reward data-efficient approaches.
○ *Reward responsible practice:* Recognise responsible resource, and transparent environmental reporting.

**Policies.** Institutions, organisations, and governments can institutionalise change by defining standards and reporting requirements and by investing in shared infrastructure.
○ *Standardise reporting:* Prescribe clear standards for reporting resource use and environmental impact due to data.
○ *Encourage shared dataset storage:* Reduce redundant local copies through shared infrastructure[13], which can host common datasets, discouraging users from storing their own copies, and enforcing efficient usage policies to discourage wasteful computations.
○ *Fund and maintain common infrastructure:* Funding agencies should support the development of shared data-related infrastructure, and mandate using them.
○ *Data usage and data sunset laws:* Using large-scale data should require justification and approval as is common when using health data[14], along with the destruction of data after a set period to reduce accumulation of dark data.

## 7. Alternative Views

**Benefits of data scaling.** The most common counter-position to data frugality advocates for continued data scaling. Proponents argue that empirical scaling laws demonstrate consistent performance gains with increasing dataset size, making further data expansion a reliable driver of progress (Kaplan et al., 2020; Hoffmann et al., 2022). From this perspective, data reduction may risk slowing innovation or prematurely constraining model capabilities.

Large datasets generally also improve robustness to distribution shift (Hendrycks et al., 2020; Liu et al., 2025) and increase coverage of rare or long-tail events that may be critical for safety and reliability (Van Horn & Perona, 2017). Consequently, data reduction can remove rare but important samples (Dharmasiri et al., 2025). In safety-critical domains, such as autonomous driving, large and diverse datasets are often necessary for capturing edge cases and ensuring robust behaviour (Lei et al., 2025; Kalra & Paddock, 2016).

---

[10]https://cvpr.thecvf.com/Conferences/2026/ComputeReporting

[11]Hosted at MICCAI https://e2mip.org/
[12]https://babylm.github.io/
[13]Like the national projects such as Sweden's Berzelius cluster.
[14]Such as UK Biobank or Human Genome Project.

Beyond technical motivations, structural and strategic incentives also sustain large-scale data practices, especially in industry. Here, proprietary datasets are a source of competitive advantage, and organisations may be reluctant to reduce data holdings for "fear of missing out" on insights and future utility (Zuboff, 2023). At frontier scales, the overhead of careful data selection can itself seem prohibitive compared to simply using all available data.

**Data-scarce domains.** In domains such as robotics, where data collection requires special hardware, expert knowledge, and significant time investment, demonstrations can be scarce and expensive to obtain. In such settings, data frugality through subset selection may be counterproductive, as removing demonstrations risks discarding rare or safety-critical behaviours that are difficult to re-collect. Data efficiency may better be pursued through other means, such as incorporating stronger inductive biases into models (Brehmer et al., 2025; Seo et al., 2025) or leveraging simulation and transfer learning. However, as robotics datasets have grown rapidly in scale (Open X-Embodiment Collaboration, 2023; Walke et al., 2023), recent work has successfully applied subset selection to imitation learning and offline reinforcement learning (Dass et al., 2026; Yang et al., 2025), indicating that data frugality is increasingly applicable even in historically data-scarce domains.

**Rebound effects.** Another counter-position holds that efficiency gains do not necessarily yield proportional reductions in overall resource use. Lower costs from more efficient methods may instead incentivise increased scale, frequency of use, or broader deployment, offsetting expected savings through rebound effects (Morand et al., 2025; Wright et al., 2025; Luccioni et al., 2025). In the context of smaller datasets, this may manifest as an increase in the number of experiments conducted rather than a net reduction in energy use. This perspective reinforces the importance of aligning efficiency gains with intentional changes in practice.

**When sustainability arguments undermine democratisation.** A further concern is that sustainability arguments may be misused to legitimise reduced access or increased centralisation. Zimmermann et al. (2025) warn that calls for restraint can unintentionally reinforce existing power asymmetries if they are formed as reasons to limit participation rather than to redesign systems. Our position explicitly rejects this interpretation: data frugality aims to expand participation by enabling meaningful experimentation under constrained resources, not to justify exclusion.

Taken together, these alternative perspectives highlight that data frugality is neither universally applicable, nor sufficient on its own, but must be situated within a broader understanding of task requirements, safety constraints, and systematic incentives.

## 8. Conclusion

Recent critiques of data scaling have successfully documented its environmental, social, and systemic costs, yet these insights have largely remained at the level of diagnosis. In this position paper, we argue that the persistent gap between *preaching and practising* data frugality is no longer justified. By making dataset-level energy and carbon costs explicit, and by demonstrating that representative subset selection can substantially reduce storage, training time, and training energy with little loss in performance, even mitigating bias, we show that data frugality is both technically feasible and practically beneficial today.

At the same time, we acknowledge that data frugality is not a universal remedy. Its effectiveness depends, e.g., on task requirements. Nevertheless, treating data as an abundant input is increasingly problematic. We therefore call for a shift in norms and evaluation practices that make data-related costs visible and reward careful data stewardship. Moving toward responsible AI development requires not only better models, but also better choices about the data we create, store, and (re-)use. Practising data frugality is a concrete step in that direction.

## Acknowledgements

SW and RS acknowledge funding from the Independent Research Fund Denmark (DFF) under grant agreement No. 4307-00143B. RS acknowledges funding from the European Union's Horizon Europe Research and Innovation Action programme (HORIZON-CL4-2021-HUMAN-01) under grant agreement No. 101070408, project SustainML (Application Aware, Life-Cycle Oriented Model-Hardware Co-Design Framework for Sustainable, Energy Efficient ML Systems). RS further acknowledges funding from the European Union's Horizon Europe Research and Innovation Action programme under grant agreements No. 101070284 and No. 101189771. GFG acknowledges funding from DTU Centre for Absolute Sustainability (CfAS). SM acknowledges funding from the Danish Data Science Academy (DDSA) through DDSA Visit Grant No. 2025-5673. AM and SM acknowledge support by the Wallenberg AI, Autonomous Systems and Software Program (WASP) funded by the Knut and Alice Wallenberg Foundation. The computations were enabled by the Berzelius resource provided by the Knut and Alice Wallenberg Foundation at the National Supercomputer Centre. The authors thank the members of SAINTS Lab for valuable discussions.

## Environmental Impact Statement

We track the energy consumption of our experiments using `carbontracker`[15] (Anthony et al., 2020). The analysis in Section 4.1 incurs negligible energy consumption, as we aggregate numbers from He et al. (2024) and Tan et al. (2025). The experiments in Section 4.2 consume 71 kWh (114 hours), corresponding to 2.56 kgCO$_2$e using the Swedish average carbon intensity of 36 gCO$_2$e/kWh for 2024 (Ember & Energy Institute, 2025). These results are reused for the experiment in Section 3.1. The experiments in Section 4.3 consume 0.24 kWh, corresponding to 34 gCO$_2$e, using the Danish average carbon intensity of 143 gCO$_2$e/kWh for 2024 (Ember & Energy Institute, 2025). Excluding the unknown cost of using ChatGPT for the ImageNet analysis (Appendix B), this position paper incurs a total of 2.59 kgCO$_2$e.

## Generative AI Usage Statement

ChatGPT version 5.1 was used to support programming tasks, including the development of scripts for accessing the openreview API and data visualisation. ChatGPT versions 5.1 and 5.2 were used to analyse the ImageNet usage of ICLR papers and to edit and refine language and grammar in selected sections of the manuscript. Google Gemini was used to identify relevant academic work.

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

# A. Overview of Data Reduction Techniques

**Coreset selection methods.** The goal of a *representative subset* is to approximate a sum over a large number of terms with fewer terms, i.e., $f(\theta) := \sum_{i=1}^{n} w_i f_i(\theta) \approx \sum_{i \in \mathcal{I}}^{n} \tilde{w}_i f_i(\theta) =: \tilde{f}(\theta)$, where $\mathcal{I}$ is a subset of the indices of much smaller size, i.e., $|\mathcal{I}| =: \tilde{n} \ll n$, and $\tilde{w}_i$ is a potentially adjusted weight. Typically, the large sum iterates over the training data and the functions $f_i$ correspond to per-point loss functions and $\tilde{f}(\theta)$ is often an unbiased approximation of the objective function $f(\theta)$. However, instead of matching the objective function, other quantities like the gradient can be matched. Without guarantees on the approximation error, the subset merely serves as a *heuristic*. When theoretical guarantees on the approximation error can be derived, we call the representative subset a *coreset*.

Numerous constructions exist for representative subsets and coresets across learning tasks, see Moser et al. (2025) for an overview. For example, constructions for $k$-means clustering (Har-Peled & Mazumdar, 2004; Lucic et al., 2016; Bachem et al., 2018) and matrix factorization (Mair et al., 2017; Mair & Brefeld, 2019) often rely on geometric principles and sensitivity scores that estimate each data sample's influence on the learning objective. Similar approaches have also been applied to logistic regression (Munteanu et al., 2018; Campbell & Beronov, 2019).

Other strategies derive informative scores from early training dynamics, such as the expected L2 norm (EL2N) or the gradient norm (GraNd) of the loss (Paul et al., 2021). These methods identify samples most critical to optimisation. For instance, GraNd enables up to 25% pruning on CIFAR-100 and 50% pruning on CIFAR-10 with negligible performance loss. This principle has also been applied to train language models using as little as 30% of the original dataset (Marion et al., 2023). Furthermore, Fayyaz et al. (2022) adapt GraNd and EL2N to NLP tasks.

Other gradient-based coreset methods aim to preserve the optimisation dynamics of training on the full dataset while enabling substantial reductions in dataset size and training time. For instance, CRAIG (Mirzasoleiman et al., 2020) selects subsets such that the gradient of the loss on the subset closely matches that of the full dataset, achieving speed-ups of up to $6\times$ for logistic regression and $3\times$ for deep neural networks. Grad-Match (Killamsetty et al., 2021) uses a similar approach, reducing dataset size to 30% for vision tasks with negligible accuracy loss. Similarly, CREST (Yang et al., 2023) demonstrates speed-ups of up to $2.5\times$ on various vision and NLP benchmarks with minimal performance loss. A combination of prediction uncertainty and training dynamics when scoring data points is approached by Dyn-Unc (He et al., 2024) while InfoMax (Tan et al., 2025) selects subsets that maximize representative information content. Both approaches demonstrate that ImageNet-1K can be pruned by up to 25% with negligible to no performance degradation. More generally, training-dynamics-based approaches also identify important examples by explicitly monitoring learning behaviour, such as forgetting events, i.e., examples that repeatedly transition from correct to incorrect classification (Toneva et al., 2019), or early-training proxy signals that predict final importance (Coleman et al., 2020). More recently, $\mathbb{D}^2$ pruning (Maharana et al.) uses a message-passing formulation that jointly balances sample interactions between data points. It improves over methods based solely on training dynamics or geometric coverage.

While all these methods reduce dataset size and accelerate training with a small drop in performance, their potential to reduce energy consumption and carbon emissions remains largely unexplored. Notable exceptions include Killamsetty et al. (2021), who quantify wall-clock speed-ups from gradient-matched subset selection as a proxy to reduce compute, and Scala et al. (2025), who explicitly evaluate data reduction strategies in terms of energy efficiency during training. Both works go beyond reporting accuracy and runtime and explicitly quantify energy savings from training on reduced datasets.

Beyond methods that explicitly construct subsets, a broader literature on data-centric metrics quantifies the importance or redundancy of individual samples and provides conceptual grounding for data frugality. Work on memorisation shows that models memorise only a small subset of atypical examples, implying that large fractions of training data may be unnecessary for good generalisation (Feldman & Zhang, 2020). Influence functions estimate the effect of individual training points on model predictions (Koh & Liang, 2017), offering a principled tool for identifying and removing low-influence samples. Such scores are closely related to the importance criteria underlying score-based pruning, such as the EL2N and GraNd scores (Paul et al., 2021) and forgetting events (Toneva et al., 2019). While these metrics are not subset-selection methods in themselves, they motivate and inform frugal data practices by making the uneven informativeness of training data explicit.

**Coreset approaches for model selection.** Several works leverage coreset and subset selection techniques to reduce the cost of model selection, including hyperparameter optimisation and neural architecture search (NAS). For example, Grad-Match (Killamsetty et al., 2021) uses gradient-matching subsets to preserve optimisation dynamics, while AU-TOMATA (Killamsetty et al., 2022) explicitly integrates subsets into the hyperparameter optimisation loop. Similar ideas have been applied to NAS, where coresets (Shim et al., 2021) and active learning approaches (Geifman & El-Yaniv, 2019)

have been used to substantially reduce model selection cost.

**Data condensation.** Dataset condensation seeks to replace large training sets with a small number of synthetic data points that induce similar training dynamics. Theoretical insights into this idea have been developed in the infinite-width regime of CNNs by Nguyen et al. (2021), while practical methods based on gradient matching made condensation effective in practice (Zhao et al., 2021). Other work has addressed scalability, enabling dataset condensation at ImageNet scale (Cui et al., 2023; Yin et al., 2023). Unlike coreset methods, which select representative subsets of data, dataset condensation produces synthetic datasets, offering often better compression at the cost of interpretability, data provenance, and additional optimisation overhead.

## B. ImageNet Usage Analysis

**ICLR-based identification of ImageNet usage.** The following section describes our analysis of ICLR papers from 2017–2022 that used ImageNet to train models from random initialisation, and how we projected these trends to subsequent years.

Using the OpenReview API, we retrieved all available papers from the version 1 (v1) endpoint. We focus on ICLR because all years except 2016 used OpenReview for peer review. For each year, we analysed accepted papers for mentions of ImageNet and recorded the corresponding forum IDs. For matched papers, we processed the full PDF using an LLM to determine the context in which ImageNet was used (e.g., training from random initialisation, fine-tuning, or citation only) and to identify the variant of ImageNet used. Our primary analysis focuses on papers that trained models on ImageNet from random initialisation.

In many cases, papers do not explicitly describe how ImageNet was used. We therefore allow the model to infer the training scenario from the surrounding text, supported by quotes to justify the classification. Similarly, the dataset variant is not always specified; for papers trained from scratch, we assume the most commonly used version is ImageNet-1K with 1.28 million training images.

To assess the reliability of the LLM-based labelling, we manually audited a random sample of 40 papers, comparing the model's assigned usage category against the ground truth determined by reading each paper. Of these, 39 were labelled correctly; the single error involved a paper that used Tiny-ImageNet for evaluation rather than training from random initialisation. This corresponds to an empirical error rate of $1/40 = 2.5\%$, which we propagate as an uncertainty band through all downstream estimates of training-run counts, energy, and carbon costs in Section 3.

As we were unable to access the 2023–2025 papers, we estimated ImageNet-related statistics for these years using ordinary least squares (OLS) linear regression fitted to the 2017–2022 trends. Table 2 reports both observed and projected values, with Figure 5 providing a visual summary. Figure 5 shows that the fractions of ICLR papers mentioning and training on ImageNet exhibit a slight decline over time; however, the total number of accepted papers has increased substantially, resulting in continued growth in the absolute number of ImageNet-using papers.

*Table 2.* Observed (2017–2022) and projected (2023–2025) ImageNet usage and training counts. Totals for 2023–2025 are real acceptance counts; ImageNet-related values for 2023–2025 are projected via linear trends fit on 2017–2022.

| | Observed | | | | | | Projected | | |
|---|---|---|---|---|---|---|---|---|---|
| Metric | 2017 | 2018 | 2019 | 2020 | 2021 | 2022 | 2023 | 2024 | 2025 |
| Total accepted (real) | 196 | 336 | 502 | 687 | 858 | 1094 | 1574 | 2260 | 3704 |
| ImageNet papers | 75 | 129 | 183 | 238 | 319 | 395 | 554 | 785 | 1270 |
| ImageNet ratio | 0.383 | 0.384 | 0.365 | 0.346 | 0.372 | 0.361 | 0.352 | 0.348 | 0.343 |
| Trained on ImageNet | 28 | 43 | 55 | 80 | 105 | 110 | 154 | 207 | 315 |
| Trained on ImageNet / total | 0.143 | 0.128 | 0.110 | 0.116 | 0.122 | 0.101 | 0.098 | 0.091 | 0.085 |

**Extrapolating ImageNet usage beyond ICLR.** To estimate total ImageNet usage beyond ICLR, we assume that the fraction of papers training on ImageNet from scratch at ICLR is representative of the broader ML literature. We therefore retrieve the total number of papers mentioning "ImageNet" between 2017 and 2025 via a keyword search on dimensions.ai[16] and apply the estimated fraction to this corpus. Table 3 summarises these estimates.

---

[16]app.dimensions.ai

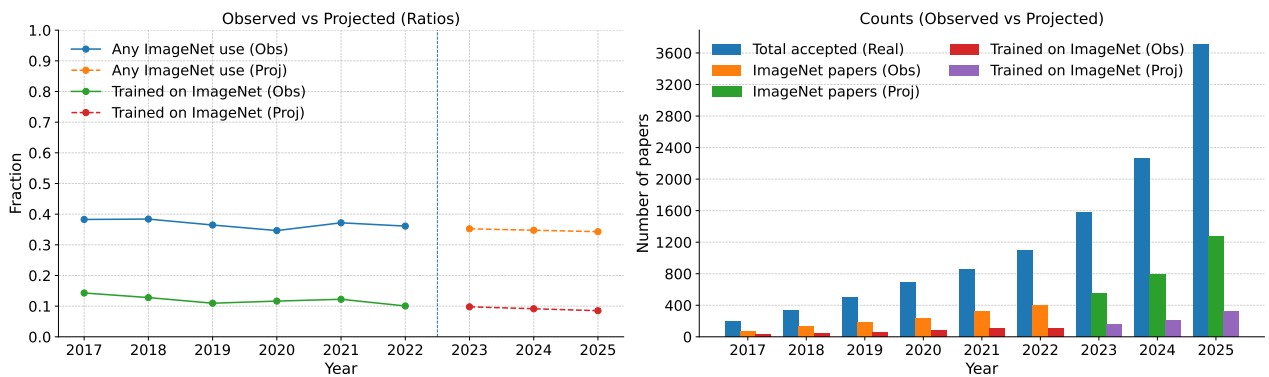

*Figure 5.* Left: Observed and projected fraction of papers at ICLR with ImageNet use. Right: Observed and projected growth of papers using ImageNet to train models from random initialization.

*Table 3.* Yearly number of publications mentioning ImageNet based on dimensions.ai and the inferred number of models trained from random initialisation, obtained by multiplying the ICLR-derived training ratio by the total number of ImageNet-mentioning papers.

|  | 2017 | 2018 | 2019 | 2020 | 2021 | 2022 | 2023 | 2024 | 2025 |
|---|---|---|---|---|---|---|---|---|---|
| Total published | 12,998 | 21,389 | 31,538 | 38,374 | 48,294 | 53,825 | 59,847 | 63,516 | 65,124 |
| Trained on ImageNet | 1,104 | 1,946 | 3,090 | 3,875 | 5,891 | 6,243 | 6,583 | 8,130 | 9,312 |

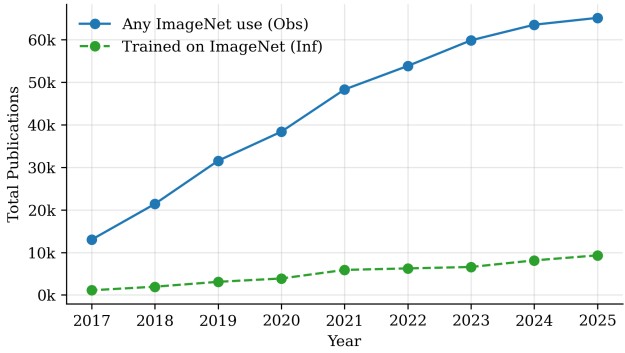

*Figure 6.* Total number of publications mentioning ImageNet (blue) retrieved from dimensions.ai and the total number of models trained on ImageNet from random initialisation (green), obtained by applying the annual ICLR-derived training ratios to the total publication counts.

Figure 6 shows the total number of publications mentioning ImageNet and the corresponding estimated number of models trained on ImageNet obtained by applying the annual ICLR-derived ratios. This procedure results in an estimated total of 46,179 ImageNet training runs over the period 2017–2025.

**Per-epoch energy measurement.** To estimate the average energy consumption per training epoch, we trained a ResNet-50 model (He et al., 2016) three times for ten epochs and tracked the energy use with carbontracker[17] (Anthony et al., 2020).

**Versions and copies stored on the Hugging Face Hub.** Since the release of ImageNet, numerous derivative versions and subsets have been released by both the original authors and the broader research community (Luccioni & Crawford, 2024). Figure 7 shows the 876 datasets hosted on the Hugging Face Hub[18] which includes *imagenet* in their dataset identifier. Among these 214 explicitly include *imagenet-1k* in their dataset identifier.

**Scope and limitations.** The estimates presented in Section 3 are strict lower bounds and reflect a deliberately narrow scope. We restrict our analysis to ImageNet-1K and to two stages of the data and model lifecycle: dataset storage and model training.

---

[17]https://github.com/saintslab/carbontracker
[18]Retrieved on 1 December 2025.

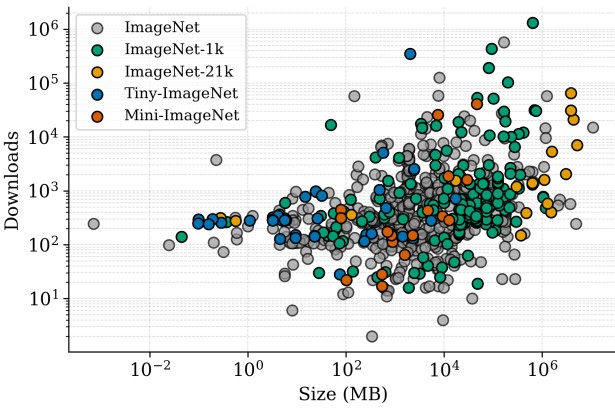

*Figure 7.* Total downloads versus dataset size (MB) for the 876 ImageNet-related datasets hosted on the Hugging Face Hub as of 1st December 2025. Among these, 214 datasets explicitly include *imagenet-1k* in their dataset identifier.

Other stages, including data collection, curation, and distribution as well as model selection, evaluation, and deployment are out of scope. We further limit our accounting to energy use, specifically electricity consumption, and associated carbon emissions, excluding broader environmental impacts such as embodied hardware emissions and end-of-life effects.

The estimated number of training runs is derived from accepted papers and therefore does not capture the many training runs that do not result in a paper. Finally, we note that ImageNet-1K is a curated subset of the larger ImageNet-21K dataset, constructed to support more tractable benchmarking while retaining broad visual coverage. Our estimates therefore capture only a subset of all ImageNet-related usage.

## C. Additional Material for Bias Experiment

Figure 8 depicts a visualisation of the data used in the experiment described in Section 4.3. Figure 9 shows the results for additional bias strengths, i.e., the fraction of samples belonging to the majority group $\{0.0, 0.75, 0.95\}$ for the following data budgets: $\{100, 500, 1000, 10000, 25000, 50000\}$. Unsurprisingly, stronger bias strength lead to larger gains from rebalancing the dataset to mitigate bias, as noted in Figure 4.

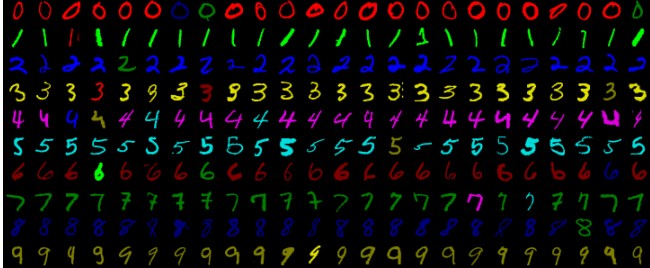

*Figure 8.* Visualisation of a batch of samples, generated with a bias probability of 0.1. Each digit has a majority group (colour) and a smaller minority of instances from other colours.

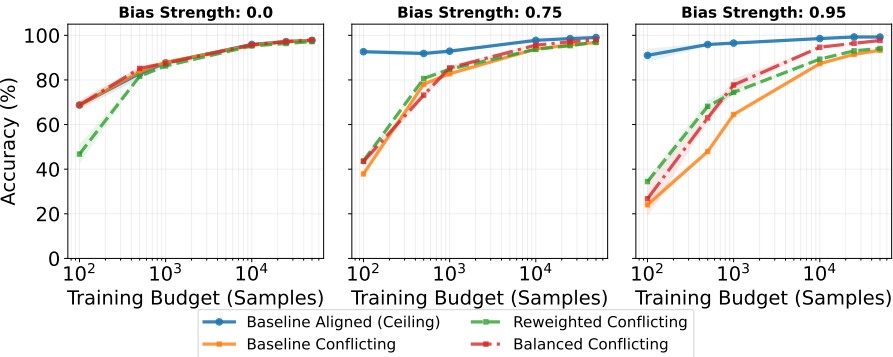

*Figure 9.* Performance of a classifier trained on the biased Colour-MNIST dataset is reported for the case with no bias (aligned) and with bias (conflicting) for four bias strengths (fraction of samples in the majority group): 0.0%, 75%, 95%, and 99%. Three coreset methods are shown: random (baseline), reweighted, and balanced.

