# OpenReview forum: "Position: Stop Preaching and Start Practising Data Frugality for Responsible Development of AI"
_ICML.cc/2026/Position_Paper_Track — ICML 2026 Position Paper Track regular_

### Official Review · Reviewer_Ryre · 2026-03-05

**Significance:** 4
**Argument Clarity:** 3
**Rating:** 5
**Confidence:** 4

**Questions:**

1. How do the authors see their position applying to robotics, imitation learning, or other sequential decision-making domains where data frugality may not work?
2. The bias mitigation experiment was performed on a very simple setup. What is the evidence for this to scale up to more realistic setups used in today's ML platforms.

**Alternative Views Section:**

Yes

**Compliance With Llm Reviewing Policy A Conservative:**

Affirmed.

**Discussion Potential:**

4

**Paper Summary:**

This paper argues that current practices rely on increasing dataset sizes instead of maximizing the learning efficiency per sample. This view is supported by energy estimates and carbon costs associated with training and storing ImageNet-1k. The paper concludes by calling on all relevant actors to make data-related costs more visible as well as reward resource-efficient practices.

**Position:**

Yes

**Position In Title:**

Yes

**Related Work:**

2

**Strengths And Weaknesses:**

## Strengths
- The paper presents the position clearly in the title and the introduction.
- The topic is relevant to all members of the ML community.

## Weaknesses
- The paper does mention that the coreset identification adds an overhead. However, since the paper presents a case for efficiency, adding a quantification (or an approximate estimation) of corest selection might be beneficial.
- The paper is very data-centric, but does not mention or discuss data-metrics such as memorization, influence scores, etc.
- Fields such as Robotics suffer more from not having enough training samples. This may provide an alternative view to the paper and is worth discussing.

**Support:**

3

---

> ### Author Rebuttal · Authors · 2026-03-30
>
> ### W1.
> We acknowledge that we do not provide an end-to-end comparison which includes the coreset construction cost. Our reasoning is principled: Our energy measurements (Sec. 4.2) are a function of dataset size rather than of the specific coreset method used, and are therefore directly applicable to any method achieving this pruning ratio. We also deliberately avoid rerunning published accuracy experiments, as doing so would itself be *wasteful and at odds with the spirit of this paper*.
>
> That said, we agree that it would strengthen the paper to include a quantification of the coreset cost. We therefore add a Subsec. 4.4, which accounts for this and its amortisation:
> > Section 4.4 Amortisation of Coreset Construction Costs
>
> > Constructing a representative subset incurs an upfront one-time cost that varies considerably across methods. At one extreme, random uniform pruning requires no construction overhead. At the other, methods such as Dyn-Unc require computation equivalent to a full training run on the entire dataset. In the latter case, adoption is only justified if the construction cost is amortised across subsequent uses, e.g., through repeated model selection runs or as a shared dataset that reduces downstream storage, distribution, and training costs for every user.
>
> > To make this concrete, Dyn-Unc’s cost is equivalent to 300 epochs on full ImageNet. Training on a 25% pruned subset saves 24-33% energy (Table 1), meaning the construction cost is amortised after three to four training runs on the coreset. Had a frugal version of ImageNet-1K been constructed and shared from the outset, the one-time construction cost would have been negligible relative to the aggregate savings. Concretely, a 25% reduction in dataset size would have saved approximately 1.4-1.9 GWh, corresponding to around 621-854 tCO$_2$e (using the global average carbon intensity), even under our conservative lower-bound assumptions.
>
> ### W2.
> We thank the reviewer for this suggestion. Memorisation metrics [1] provide evidence that large fractions of training data may not be necessary for good generalisation, as models only “memorise” a small subset of atypical examples. This directly supports our data frugality position. Influence scores [2] similarly provide data valuation tools that could be used to identify and remove low-influence samples. We add a brief discussion of these related data-centric metrics in Sec. 2 or App. A to situate our work more broadly within the data-centric ML literature.
>
> References:
>
> [1] Feldman and Zhang. "What neural networks memorize and why." NeurIPS, 2020.
>
> [2] Koh and Liang. "Understanding black-box predictions via influence functions." ICML, 2017.
>
> ### W3/Q1.
> The reviewer raises an important nuance. In settings where robot demonstrations are expensive to collect, data scarcity is a genuine constraint and removing demonstrations would be counterproductive. However, this framing is increasingly outdated for the field as a whole. Modern robotics datasets such as [3,4] now contain tens of thousands of trajectories across diverse embodiments and tasks, and the question of which demonstrations to select for a specific downstream task is an active research problem closely analogous to coreset selection. Recent work, e.g., [5, 6], explicitly applies influence-score and submodular subset selection methods to imitation learning and offline RL, demonstrating that data frugality does apply in these domains, with genuine benefits. We add a paragraph to Sec. 7 acknowledging the genuine data-scarcity concern in robotics while noting that the field is increasingly operating in a regime where data selection matters as much as data collection.
>
> References:
>
> [3] Vuong et al. "Open x-embodiment" Workshop @ CoRL, 2023.
>
> [4] Walke et al. "Bridgedata v2: A dataset for robot learning at scale." CoRL, 2023.
>
> [5] Dass et al. "DataMIL: Selecting Data for Robot Imitation Learning with Datamodels." Workshop @ CoRL, 2025.
>
> [6] Yang et al. "Fewer May Be Better: Enhancing Offline Reinforcement Learning with Reduced Dataset." ICLR, 2025.
>
> ### Q2.
> Adjusting the data distribution is a well-established approach to alleviating bias and improving fairness of ML models, typically by reweighting samples from minority groups, or subsampling samples from majority groups [7,8]. Several recent works have attempted to use coreset methods and other data frugality approaches to simultaneously improve accuracy and fairness trade-off for more complex datasets [9,10]. This further strengthens the empirical evidence we provided and we include these references in the manuscript.
>
> References:
>
> [7] Kamiran et al. "Data preprocessing techniques for classification without discrimination." KaIS, 2012
>
> [8] Krasanakis et al. "Adaptive sensitive reweighting to mitigate bias in fairness-aware classification." WWW, 2018
>
> [9] Zhang et al. "Fair Bayesian Data Selection via Generalized Discrepancy Measures." AAAI, 2026
>
> [10] Zhou et al. "FairDD." NeurIPS, 2025

---

> > ### Author Rebuttal · Reviewer_Ryre · 2026-04-01
> >
> > The authors have addressed my concerns. I think the addition of the discussions the authors have posted strengthens the paper. I will revise my score to reflect this.

---

### Official Review · Reviewer_T27B · 2026-03-11

**Significance:** 4
**Argument Clarity:** 3
**Rating:** 4
**Confidence:** 3

**Questions:**

1. Can the authors provide the exact list of the ten coreset papers surveyed in Section 1, along with the criteria used to determine whether they reported time or energy?
2. How was the LLM-based labeling of ImageNet usage in Appendix B validated? Was any manual audit performed?
3. Can the authors provide uncertainty bounds or sensitivity analyses for the ImageNet energy/carbon estimates under alternative assumptions?
4. Can the authors add one direct end-to-end comparison for a specific coreset method, including subset-construction cost and training-to-convergence energy?

**Alternative Views Section:**

Yes

**Compliance With Llm Reviewing Policy A Conservative:**

Affirmed.

**Discussion Potential:**

4

**Final Justification:**

Most of my concerns have been resolved, but there is not enough evidence to boost my score. I maintain my borderline accept score.

**Paper Summary:**

The study's broad domain is sustainable and responsible machine learning. The authors attempt to discuss a significant question: how the ML community can move from rhetorical support for data frugality to actual practice.

This position paper argues that continued dataset scaling has diminishing returns and under-accounted environmental costs, and that the community should more actively adopt data-frugal approaches. To support this position, the paper: (i) reviews background on environmental costs of data and data-frugal learning, especially coreset selection; (ii) provides lower-bound estimates of the downstream training and storage energy/carbon associated with ImageNet-1K usage; (iii) summarizes prior ImageNet pruning results suggesting that moderate pruning can preserve accuracy; (iv) measures training-time and energy reductions from training on smaller ImageNet subsets; (v) uses a toy Colored-MNIST example to argue that curated subsets can also help mitigate bias; and (vi) proposes recommendations for researchers, platforms, and policy makers.

**Position:**

Yes

**Position In Title:**

Yes

**Related Work:**

4

**Strengths And Weaknesses:**

**Strengths**
1. Highly relevant to ICML. The topic is timely and important for the ICML community, touching sustainability, data-centric ML, benchmark culture, reporting norms, and accessibility of research.
2. Clear central position. The paper has a clear thesis—closing the gap between “preaching” and “practising” data frugality—and the recommendations are concrete rather than purely rhetorical.
3. Good breadth of related work. The paper connects technical coreset/data-pruning literature with sustainability, governance, and recent position papers in a useful way.
4. Constructive attempt to quantify the issue. Section 3 is valuable because it tries to make dataset-level environmental costs concrete rather than leaving them abstract.

**Weaknesses**
1. Evidence is narrower than the scope of the claim. The paper makes a broad argument about responsible AI development and even frontier AI, but most concrete evidence comes from ImageNet classification plus a toy Colored-MNIST bias example. The empirical base does not yet fully support the breadth of the position.
2. Section 3 relies on strong assumptions. The ImageNet cost estimate depends on several compounding assumptions: ICLR trends generalize to all ML literature, paper counts approximate training runs, LLM-based labeling is accurate, a single ResNet-50/A100/300-epoch setup is representative, and one local copy per run is a reasonable storage model. As written, this is useful as an illustrative back-of-the-envelope estimate, but not as a robust measurement.
3. No uncertainty or sensitivity analysis. Given the assumptions above, the paper would be much stronger with uncertainty ranges or scenario analysis.
4. End-to-end support is incomplete. The paper combines accuracy evidence from prior work with its own training-energy measurements on random 25% pruning. That is suggestive, but it is not the same as an end-to-end comparison of a specific coreset method, including subset-construction cost and training-to-convergence energy.

**Support:**

3

---

> ### Author Rebuttal · Authors · 2026-03-30
>
> ### W1.
> We acknowledge this critique, but believe ImageNet is a representative choice: it has been one of the most widely used benchmarks for over 15 years, and its accumulated energy and carbon cost (Sec. 3) makes it a compelling case study. Demonstrating that 25% of this data could have been pruned from the outset with negligible accuracy loss (Sec. 4.1), and quantifying the energy saving (Sec. 4.2), carries a lesson that extends beyond image classification: the frugal dataset practices we advocate for should inform how datasets, incl. tokenised, multimodal, and ones used for frontier AI, are developed going forward. In this way, our position and arguments extend to responsible AI development more generally.
>
> That said, we acknowledge that our empirical findings may not generalise directly to tokenised/multimodal datasets used in frontier models (L372-374). However, these types of datasets have also shown to contain substantial redundancy (Sec. 2.3), and we therefore argue that data frugality extends to frontier datasets even when our experiments do not directly demonstrate this.
>
> ### W2.
> We are pleased that the reviewer finds the section valuable despite its limitations. As we note in L242-243, our estimates are strictly lower bounds. Exact measurement is infeasible in the absence of available metadata on training runs and storage practices at scale. In our opinion, a carefully constructed back-of-the-envelope estimate is the best approach available. We agree this necessarily involves strong assumptions, but we would argue that this is preferable to not accounting for these costs at all. To our knowledge, this is the first attempt to quantify the aggregate downstream energy use and carbon footprint of a widely used benchmark dataset. We believe this is a valuable first step towards the kind of dataset-level accounting that the community should be working towards.
>
> ### Q2.
> We initially verified the LLM output by manually checking ten random papers, which were all correct. However, we repeated this validation to a random sample of 40 papers of which 39 were correct. The one incorrect case involved a paper that used Tiny-ImageNet for evaluation, not pre-training. Thus, the error rate is 1/40=2.5%.
>
> ### W3/Q3.
> We thank the reviewer for this suggestion and add sensitivity analyses along two dimensions:
>
> (i) We propagate the LLM output uncertainty (2.5%, see Q2) through all relevant estimates, including the number of training runs and the energy and carbon costs for both training and storage.
>
> (ii) We include low and high carbon intensities to address the variation in carbon intensity across energy grids by adding the following (after L240):
>
> > Carbon footprint estimates are highly sensitive to the carbon intensity of the local energy grid, which varies by more than a factor 60 across countries [1]. To illustrate this sensitivity, we convert the total energy consumption of 5.82 GWh to carbon footprints under three scenarios: using the global average (445 gCO$_2$e/kWh), a high-carbon grid such as Turkmenistan (1,310 gCO$_2$e/kWh), and a low-carbon grid such as Lesotho (21 gCO$_2$e/kWh). This yields estimates of 2,589 $\pm$ 65 tCO$_2$e, 7,624 $\pm$ 191 tCO$_2$e, and 122 $\pm$ 3 tCO$_2$e, respectively.
>
> References:
>
> [1] https://archive.ourworldindata.org/20251014-145858/grapher/carbon-intensity-electricity.html
>
> ### W4/Q4.
> We acknowledge that combining accuracy evidence from prior work with our own energy measurements is not the same as a full end-to-end comparison. Our reasoning for doing so is principled rather than a shortcut. Our energy measurements (Sec. 4.2) are a function of dataset size rather than of a specific coreset method, and are therefore directly applicable to any method which achieves this pruning ratio. We also deliberately avoid rerunning accuracy experiments that have already been published, as doing so would itself be wasteful and at odds with the spirit of this paper.
>
> We agree that quantifying and discussing the construction cost is important and adds value to the paper. We therefore add a Subsec. 4.4, which accounts for coreset construction costs and their amortisation across downstream use, **see response to Reviewer Ryre W1**.
>
> ### Q1.
> *Ten methods:* AdaCore [2], CRAIG, CREST, D2, DynUnc, Entropy [3], GLISTER [4], GRAD-MATCH, GraNd, InfoMax
>
> *Motivations:*
>
> Comp. efficiency: AdaCore, CRAIG, CREST, DynUnc, GLISTER, GRAD-MATCH, GraNd, InfoMax
>
> Energy efficiency: CRAIG, CREST, GRAD-MATCH
>
> Storage savings: DynUnc, InfoMax
>
> Democratisation of AI: GraNd, DynUnc
>
> *Reported evaluations:*
>
> Time savings: AdaCore, CRAIG, CREST, Entropy, GLISTER, GRAD-MATCH
>
> Energy savings: GRAD-MATCH
>
> References:
>
> [2] Pooladzandi et al. "Adaptive second order coresets for data-efficient machine learning." ICML, 2022.
>
> [3] Coleman et al. "Selection via Proxy." ICLR, 2020.
>
> [4] Killamsetty et al. "Glister." AAAI, 2021.

---

> > ### Author Rebuttal · Reviewer_T27B · 2026-04-01
> >
> > Thanks for your time and rebuttal.
> >
> > The manual audit of the LLM-based labeling and the explicit list of surveyed methods improve transparency. However, my  concerns remain partially addressed: (1) the evidence base remains narrower than the breadth of the position, (2) the added sensitivity analysis does not address the dominant assumptions, (3) there is still no direct end-to-end comparison of a specific coreset method including subset-construction cost.
> >
> > Therefore, I keep my score, which is already positive.

---

### Official Review · Reviewer_xDkq · 2026-03-12

**Significance:** 1
**Argument Clarity:** 2
**Rating:** 2
**Confidence:** 5

**Questions:**

Please refer to the section above.

**Alternative Views Section:**

Yes

**Compliance With Llm Reviewing Policy A Conservative:**

Affirmed.

**Discussion Potential:**

1

**Final Justification:**

There has been tremendous effort in the community for data efficiency through various technologies such as data pruning, coreset, data distillation, data condensation, etc. This paper disregards such efforts that have been in the community for a long while. The community has been "practicing", unlike the authors' claim. Hence, the claim is wrong (it is not about agreeing or disagreeing with it), and I maintain my score of Reject.

**Paper Summary:**

This paper argues that the gap between preaching data frugality and practicing it must be closed. The authors argue that coreset-based subset selection can substantially reduce training energy consumption with little loss in accuracy.

**Position:**

Yes

**Position In Title:**

Yes

**Related Work:**

2

**Strengths And Weaknesses:**

It is not significantly convincing to argue to close the gap between “preaching and practicing.” The community and the researchers are already aware that we ought to use data sparingly. The position/argument is not new, and even such efforts have been devoted.

However, why does it have to be core-set selection? In order to obtain a coreset, one needs to wrestle with the entire dataset. Then it would matter when the coreset will be used, and then the question is the cost for (obtaining a coreset + training with coreset) < (training with the entire dataset). Although the authors mentioned this issue in paragraphs (lines 345-366 on page 7, and as an Alternative view through lines 417-427 on page 8), they did not provide any measurement or analysis on that. I would have liked the paper if they showed it , and also showing it would have strengthened their argument – although this is a position paper, such supporting details are expected.

If I am being fully honest, what additional value does this paper provide that the early/original papers that addressed coreset work (e.g., Feldman 2019)? In some sense, this paper ultimately sells a coreset idea.

Although this paper mix up data pruning and coreset selection to some extent, they are different. In particular, for example, the authors showed a case of a data set where 25%-50% is pruned. I do not believe this is a case of coreset selection – “sampling uniformly at random” is not a coreset selection. A coreset selection, as opposed to data pruning, is supposed to obtain a much smaller representative subset (much fewer samples.)

In one of their discussions in section 5 (the paragraph, Alternative to data frugality), the authors suggested efficiency gains at the level of models or hardware. I strongly disagree with this point. Such efforts to obtain efficient models may have a higher impact on resource/energy footprints, so it will return an opposite consequence than the one the authors desired.

**Support:**

2

---

> ### Author Rebuttal · Authors · 2026-03-30
>
> ### Position is not new, such efforts have already been devoted. + Additional value over earlier work on coresets (e.g. Feldman 2019).
>
> The reviewer writes that *“The position/argument is not new, and even such efforts have been devoted”*, without providing any references to support their claims. The reviewer later provides a reference when asking *“What additional value does this paper provide that the early/original papers that addressed coreset work (e.g., Feldman 2019)?”* This study is a seven-year-old survey on coreset methods, that addresses none of the specific contributions we make: identifying the preaching-practising gap, estimating energy and carbon costs at the dataset level, quantifying energy reduction or bias mitigation achievable through coreset methods, or providing actionable recommendations for multiple stakeholders. It therefore does not support the claim that our contribution has already been made.
>
> We also note that novelty is not the primary criterion for a position paper; relevance, evidence, and actionability are. But more fundamentally, even granting the existence of prior advocacy strengthens rather than undermines our position, as, despite such calls, data scaling continues to dominate practice, which is precisely the gap we point out.
>
> Finally, the reviewer's own comment undermines their critique. They write that *“the community and researchers are already aware that we ought to use data sparingly”*, yet it is unclear whether this refers to the data frugality community specifically or to the broader ML community. If the former, the observation is trivially true, but even within this community a value-action gap persists, as our observations in Sec. 1 demonstrate. If the latter, we disagree with the belief that the broader ML community routinely considers adopting data frugal approaches, which points to a lack of awareness rather than mere inaction. Either interpretation strengthens rather than weakens the case for our position.
>
>
> ### Why focus on coresets? + Distinction between pruning and coreset selection.
>
> We acknowledge that our paper focuses primarily on subset selection and coreset methods as a means of achieving data frugality. We now add a sentence to the manuscript clarifying that data frugality can be achieved through several approaches, such as data condensation and distillation, and that this paper focuses on subset-based methods.
>
> Regarding the distinction between data pruning and coreset selection, we adopted the terminology from the papers we consider in Section 4.1, e.g., D2, Dyn-Unc, and InfoMax, which use the terms interchangeably (see *“Dataset pruning (or coreset selection)”* in Dyn-Unc). As for the reviewer’s claim that *“‘sampling uniformly at random’ is not a coreset selection”*, we note that the Feldman (2019) [1] survey cited by the reviewer states in Sec. 2.4.3 that a *“uniform sample from the input is probably the most common ‘coreset’”*. Uniform sampling is indeed a valid coreset construction within the sensitivity framework [2] and comes with (poor) theoretical guarantees and requires many samples to be practical. Nevertheless, uniform sampling serves as an important baseline and our Fig. 3 shows that it does not necessarily perform bad. This also speaks directly to the reviewer’s concern about the overhead of adopting data frugal methods: even the most naive approach, requiring no preprocessing or wrestling with the dataset, can be sufficient to yield meaningful energy savings.
>
> References:
>
> [1] Feldman, D. Core-sets: Updated survey. In Sampling Techniques for Supervised or Unsupervised Tasks. Springer, 2019.
>
> [2] Feldman and Langberg. "A unified framework for approximating and clustering data." STOC, 2011.
>
>
>
> ### Missing cost of constructing a coreset.
>
> We agree that quantifying and discussing the construction cost is important and adds value to the paper. We therefore add a Subsec. 4.4, which accounts for coreset construction costs and their amortisation across downstream use. **Kindly see our response to reviewer Ryre W1**.
>
>
> ### Efficiency gains and rebounds effect at the level of models and hardware.
> The reviewer points to the rebound effect at the level of models and hardware. We wish to clarify that model and hardware efficiency is presented as alternative views to data frugality, not as something this paper advocates for. We agree that the rebound effect is a serious challenge, which is why we explicitly discuss it in L417-425. Importantly, the rebound effect applies equally to efficiency gains at the level of models, hardware and even data. This is why we argue that efficiency gains must be aligned with intentional changes in practice (L425-427), which in turn aligns with our broader position: technical solutions alone are insufficient, and closing the value-action gap requires deliberate changes in norms, reporting, and incentives (Sec. 6).

---

> > ### Author Rebuttal · Reviewer_xDkq · 2026-04-02
> >
> > The authors did not need to be pointed to any specific papers about coreset, because the notion of coreset matters not specific papers.
> > I do not believe the authors had an accurate and precise distinction between data pruning and coreset approach - and now I see that was why it was described in the paper in a confusing way. Data pruning is different from coreset approach, and I do not have what I stated in my review, but simply speaking, the outcome (cases) of data pruning may include the outcome of coreset approach, but not the other way around, because they are different.
> >
> > Now, the authors changed their claim from coreset to adding data condensation and distillation. Then now the authors need to show the overhead cost to construct data condensation and distillation. Having said that, my main concern still remains - this paper does not add any value on top of data efficiency. Researchers have been using data pruning, coreset, data distillation, and data condensation, because of data efficiency, the purpose of which is what this paper is trying to argue about.
> >
> > As for the last point, although the authors argued that it is an independent matter which is out of the scope of the paper, they argued the model efficiency in the submission, which is in conflict with itself, and which is absolutely incorrect.
> >
> > For these reasons, I maintain my score.

---

### Official Review · Reviewer_HS6e · 2026-03-13

**Significance:** 3
**Argument Clarity:** 2
**Rating:** 4
**Confidence:** 4

**Questions:**

See Weaknesses

**Alternative Views Section:**

Yes

**Compliance With Llm Reviewing Policy A Conservative:**

Affirmed.

**Discussion Potential:**

2

**Final Justification:**

I appreciate that the authors' efforts during the rebuttal phase and the answers do tend to address many concerns. However, overall, I believe that the paper would benefit a lot by modifications to address some other issues raised in the review. So, I will maintain my current score of weak accept.

**Paper Summary:**

The paper strongly urges practitioners to incorporate data frugality in the development of AI. Here data frugality refers to data subset selection so that uninformative or redundant samples are removed. There is a reasonably large amount of work on coreset selection (Data subsets with guarantees) for ML. The works often refer to computational and time efficiency advantages as well reduction in carbon footprint of models when trained using coresets. However, there has not been an equivalent acceptance of this method in practice. The paper tries to argue in favour of data frugality by providing empirical evidence of effectiveness of coresets in reducing training energy while retaining comparable accuracy. In addition, the paper also shows how coresets can help mitigating data bias.

**Position:**

Yes

**Position In Title:**

Yes

**Related Work:**

2

**Strengths And Weaknesses:**

Strengths:
1. The paper argues about a problem of importance to the broad ML community and hence will be of significant interest.
2. The paper is well written and mostly easy to follow.
3. The paper puts up a strong argument supporting its case by explicitly quantifying the energy and carbon costs.

Weaknesses
1. The alternative opinion section can still be improved. The paper can try to highlight the perspective as to why most industries would still rely on scale through huge data for development of AI.
2. The claim of effectiveness of data frugality in reducing bias should be supported by more evidence. It is not clear if coresets can achieve accuracy and better fairness simultaneously.

**Support:**

3

---

> ### Author Rebuttal · Authors · 2026-03-30
>
> ### W1.
>
> We appreciate the suggestion and strengthen our alternative views section in two ways. First, we follow the reviewer's suggestion and add the following paragraph to the “benefits of data scaling” paragraph (L399-416):
>
> > Beyond technical motivations, structural and strategic incentives also sustain large-scale data practices, especially in industry. Here, proprietary datasets are a source of competitive advantage, and organisations may be reluctant to reduce data holdings for ''fear of missing out'' on insights and future utility [1]. At frontier scales, the overhead of careful data selection can itself seem prohibitive compared to simply using all available data.
>
> We note that while these perspectives related to industry are real, our position paper mainly addresses the academic ML community, where structural and strategic motivations are less present.
>
> Second, we add a paragraph, where we discuss how our position relates to a domain such as robotics, which historically has been data-scarce. Here other approaches such as incorporating inductive biases into the models, may be more suitable for increasing data efficiency. **Kindly see our response to reviewer Ryre in W3/Q1 for more details.**
>
> References:
>
> [1] Zuboff. “The age of surveillance capitalism.” Social theory re-wired. Routledge, 2023.
>
> ### W2.
>
> We acknowledge that our experiment does not constitute an explicit simultaneous accuracy-fairness evaluation. However, it does suggest that both can be achieved together.
> Specifically, bias-aware coreset curation through re-weighing achieves substantially higher accuracy compared to random sampling on the conflicted test set at lower data budgets, while approaching the accuracy of a model trained on a fully bias-free dataset as the data budget increases. This suggests that bias-aware coreset curation can mitigate bias while simultaneously improving overall accuracy, even if this is not evaluated explicitly.
>
> To directly support the simultaneous accuracy-fairness claim, we now reference existing work on fair coreset selection. **Please see our response to reviewer Ryre Q2 for more details.**

---

> > ### Author Rebuttal · Reviewer_HS6e · 2026-04-02
> >
> > I agree with reviewer T27B on points 1 and 3 in the rebuttal acknowledgement and I am currently keeping my score which is already positive

---

### Decision · Program_Chairs · 2026-04-30

**Decision:**

Accept (regular)

**Comment:**

The paper identifies an important and timely socio-technical gap in the ML community, and offers support for their position, that the community should systematically implement data frugality practices, via conceptual arguments as well as empirical evidence, by approximating energy and carbon costs on the ImageNet dataset. The paper is well written and easy to follow and I believe carries an important message that is likely to spark productive discussions in the community.

The reviewers also raised a number of concerns, primarily (i) the paper does not account for the cost of actually creating the coreset; (ii) incorrect terminology used, failing to distinguish between coreset and data pruning; (iii) pushback against the claim that the community isn’t “practicing”, as there is a lot of research in these areas, (iv) claims are broader than what the results can support; e.g. results only on ImageNet, and under some specific assumptions, and more evidence needed to support the claim of bias reduction from data frugality.

During the rebuttal, the authors were able to address most of the reviewers’ concerns. (i) They add a section (whose contents are shown in the rebuttal) that quantifies the (one-off) coreset construction cost, and discuss its amortization. They also discuss that the baseline of random pruning does not entail any cost; (ii) They justify how they use those terms, showing references and quotes from established works in the literature that use the terms in the same way they do;  (iii) clarify that they don’t disagree that there is a lot of research in pruning, coresets, and related areas, but those don’t explicitly address the gap between preaching and practicing, nor discuss how and why the community should establish these practices systematically; (iv) back up claims better by adding references and discuss limitations of not using larger datasets in their experiments.

Most reviewers signaled that most of their concerns have been addressed and recommend acceptance. The outstanding issue seems to be the novelty of the position issue that Reviewer xDkq raised, who points out that the community has already been putting in a lot of effort in these research directions, hence already “practising”. However, I agree with the authors that the nature and scope of their contribution is different from what is presented in prior research papers, e.g. on coresets. I believe the paper makes a valuable call to action to the community and can stimulate important discussions and therefore recommend acceptance.